# Evaluating large language models for diagnostic reasoning from unstructured clinical narratives in epilepsy

Meghal Dani [1,2,3] ✉, Muthu Jeyanthi Prakash [1,3,4], Filip Rosa[4,5], Zeynep Akata[6,7,8] & Stefanie Liebe[3,4,5] ✉

## Abstract

**Background** Large Language Models (LLMs) have been shown to encode clinical knowledge. Many evaluations, however, rely on structured question-answer benchmarks, overlooking critical challenges of interpreting and reasoning about unstructured clinical narratives in real-world settings.

**Methods** In this study we task eight Large Language models including two medical models (GPT-3.5, GPT-4, Mixtral-8 × 7B, Qwen-72B, LlaMa2, LlaMa3, OpenBioLLM, Med42) with a core diagnostic task in epilepsy: mapping seizure description phrases-after targeted filtering and standardization-to one of seven possible seizure onset zones using likelihood estimates. We conduct quantitative and qualitative analyses, measuring correctness, confidence, calibration, and expert-evaluated reasoning quality and source citation accuracy. Through systematic prompt-engineering and ablation studies, we assess how model performance depends on variations in prompt strategy, clinical role impersonation, narrative length, and language context.

**Results** Most models yield well-above chance accuracy after prompt engineering that even approaches clinician-level performance. Specifically, clinician-guided chain-of-thought reasoning leads to the most consistent improvements. Performance is further strongly modulated by clinical in-context impersonation, narrative length and language context (13.7%, 32.7% and 14.2% performance variation, respectively). However, reasoning analysis by clinical experts reveal that correct prediction can be based on hallucinated knowledge and inaccurate source citation, underscoring the need to improve interpretability of LLMs in clinical use.

**Conclusions** Overall, *SemioLLM* provides a scalable, domain-adaptable framework for evaluating LLMs in clinical disciplines where unstructured verbal descriptions encode diagnostic information. By identifying both the strengths and limitations of LLMs, our work contributes to testing the applicability of foundational AI systems for healthcare.

## Plain Language Summary

Large language models (LLMs) are increasingly used in healthcare, but their reliability with real patient descriptions remains unclear. This study developed a framework to evaluate eight LLMs on a core epilepsy diagnostic task: determining which brain region causes seizures based on patient descriptions, crucial for treatment management. We assess correctness, confidence, and reasoning quality across 1200 seizure descriptions. Most models performed well above chance, sometimes approaching clinician-level accuracy. Performance was strongly influenced by clinician-written reasoning examples, clinical role context, description length, and language. However, some correct answers relied on fabricated facts and flawed reasoning, raising safety concerns. These findings reveal both potential and limitations of LLMs in healthcare, highlighting the need for careful validation before clinical deployment.

Large Language Models (LLMs) have shown significant potential in leveraging clinical knowledge on structured question answering (Q&A) datasets such as MedQA[1], PubMedQA[2], MedMCQA[3] and BioASQ-QA[4] in multiple medical domains[5–9]. While Q&A set-ups are advantageous as they provide a clear ground-truth for model testing, they oversimplify clinical decision-making [10], which often relies on extracting crucial diagnostic information from unstructured patient interviews containing complex, irrelevant and everyday language[11–13]. While LLMs have shown remarkable capabilities in extracting meaningful information from unstructured text for a wide range of downstream tasks in other domains[14], their ability to do so in clinical contexts remains poorly understood. Thus, clinical narratives from patients provide crucial diagnostic information to be leveraged by LLMs especially

[1]University of Tübingen, Tübingen, Germany. [2]Machine Learning in Science, Excellence Cluster Machine Learning, Tübingen, Germany. [3]Hertie Institute for AI in Brain Health (Hertie AI), Tübingen, Germany. [4]Dept. of Neurology and Epileptology, University Clinic Tübingen, Tübingen, Germany. [5]Hertie Institute for Clinical Brain Research, Tübingen, Germany. [6]Technical University of Munich, Munich, Germany. [7]Helmholtz Munich, Munich, Germany. [8]Munich Center for Machine Learning, Munich, Germany. ✉e-mail: meghal.dani@uni-tuebingen.de; stefanie.liebe@uni-tuebingen.de

when structured input is limited or absent [9,15]. However, it is an open and largely unexplored question of how well LLMs can extract and interpret clinically meaningful information from unstructured clinical narratives to support real-world diagnostic reasoning.

Neurological disorders, such as epilepsy, provide a particularly compelling use case to explore this question, as behavioral and sensory symptoms can often be directly linked to underlying brain pathologies. In epilepsy, clinicians routinely base clinical decisions on patient and witness accounts of seizure manifestations-known as *semiology*[16,17]. Especially during early diagnostic evaluations, correctly interpreting seizure symptoms is crucial for guiding follow-up procedures such as brain imaging, EEG and surgical planning [18,19]. Seizure descriptions contain diagnostically relevant information about the seizure origin in the brain, and help classifying seizure types and syndromes. For example, repetitive chewing, swallowing, or lip-smacking strongly indicate temporal lobe involvement[20], while excessive limb movements or pelvic thrusting are indicative of frontal lobe seizures[19]. Accurate localization of the seizure onset zone (SOZ) is particularly important for patients with drug-resistant epilepsy, where surgical resection of the SOZ remains the only potentially curative treatment option[20,21].

In this study, we develop a structured and automated evaluation framework, *SemioLLM*, that benchmarks LLMs' ability to extract and translate diagnostically relevant information from seizure descriptions into probabilistic seizure locations in the brain. Using an annotated database linking over 1,200 seizure descriptions to seizure foci[22], we evaluate eight LLMs, including proprietary and open-source models (GPT-3.5[23], GPT-4[24], Mixtral-8 × 7B[25], Qwen-72B[26], LlaMa2 70B[27], and LlaMa3 70B[28], OpenBioLLM, Med42[29]). We systematically analyze model accuracy, confidence, calibration and reasoning, benchmarking their outputs in comparison to evaluation by a clinical domain expert.

Our study reveals several key insights: (i) We demonstrate that LLMs can predict seizure onset zones (SOZ) based on unstructured seizure descriptions, significantly outperforming chance-level predictions. Importantly, with refined prompting techniques, for example chain-of-thought (CoT) reasoning, their accuracy substantially improves and approaches clinician-level performance. (ii) Many models exhibit reasonable trustworthiness, as assessed through an entropy-based confidence measure, where confidence similarly improves with prompt-engineering. Notably, GPT-4 and Mixtral-8 × 7B demonstrated an optimal balance of accuracy *and* confidence. (iii) Through an extensive manual reasoning assessment conducted by a domain specific clinical expert, we identify that GPT-4 demonstrates superior capabilities in integrating domain knowledge, clinical inference, and evidence verification. Mixtral-8 × 7B, while competitive in text comprehension, exhibits notable limitations in reasoning and accurate source citation, underscoring areas for future model refinement. (iv) Our analysis identifies key factors that influence LLM diagnostic performance in processing seizure descriptions. First, we observe a U-shaped relationship between description length and accuracy, with both very short and highly detailed narratives yielding better results than those of moderate length. Second, prompting models to impersonate clinical experts markedly improves both accuracy and confidence, suggesting better alignment with domain-specific reasoning. Third, our multilingual evaluation shows that while English-trained models perform well even when processing non-English clinical narratives, accuracy declines for different prompt languages, underscoring the need for targeted multilingual user adaptation.

In summary, our study provides a systematic and in-depth investigation of LLMs in epilepsy diagnostics based solely on verbal symptom descriptions, identifying prompt strategies and expert impersonation as the most significant factors that improved diagnostic accuracy on average by 10% and 13.8%, respectively. Unlike existing structured Q&A evaluations, SemioLLM demonstrates how LLMs can translate unstructured clinical narratives into probabilistic diagnostic decisions, and identifies conditions under which both accuracy and confidence can be improved. Our framework provides a practical guideline for similar deployments and is easily transferable to other clinical specialties where symptom descriptions inform diagnostic decisions, potentially improving early diagnosis and treatment planning for patients with complex neurological and other medical conditions.

## Methods

### Dataset and Preprocessing

In this study, we use the publicly available dataset, Semio2Brain[22], which systematically maps seizure semiologies to brain regions through a comprehensive meta-analysis that includes 4643 patient data constructed using PRISMA guidelines[30] from 309 peer-reviewed publications. The dataset contains localizing data points that represent the number of patients exhibiting a reported semiology with seizure onset zones (SOZ). In particular, Semio2Brain contains 35 ictal semiological categories[18], as well as a postictal and asymptomatic category, providing the most extensive public coverage of seizure manifestations to date[22]. Each semiology is linked to one or more of 103 unique brain regions, organized within seven major areas: temporal lobe, frontal lobe, cingulate gyrus, parietal lobe, occipital lobe, insula, and hypothalamus.

Each entry in the database includes a description of a seizure symptom, either a behavioral or sensory observation during a seizure, and is assigned to one or more of the seven major brain regions. The assignment of brain regions to seizure descriptions is based on two types of information: (i) Postoperative Seizure Freedom: Knowledge about seizure freedom after resection of the brain region and (ii) simultaneously Recorded Seizure Activity: Seizure patterns recorded from intracranial or surface-based EEG located within the brain region. Both types of information serve as potential ground truths linking seizure semiology to SOZ in clinical practice.

For our task, we focus on cases based on post-operative seizure freedom, as this is considered the gold standard for post hoc evaluation of successful SOZ identification. To effectively leverage the Semio2Brain dataset for our task, we perform several data preprocessing steps, including expanding abbreviations in the semiology descriptions, correcting spelling errors, and removing uninformative words and keywords (Supplementary Fig. S1). This refinement process resulted in a final dataset of 1,269 reported semiology entries, each linked to one or more of the seven major brain regions for SOZ localization tasks.

### Task Formulation

Mathematically, we task a pretrained LLM to estimate a structured likelihood distribution L across predefined brain regions given seizure semiology. Specifically, the input prompt $\widehat{P}$ comprises persona P, the user query Q, and the instruction format I for a given semiology S. And we obtain a dictionary output D such that:

$$D = r : L(r|S, P, Q, I) \, \forall r \in R \qquad (1)$$

where r is the key and likelihood is the value and R = {"Temporal Lobe", "Frontal Lobe", "Cingulate Gyrus", "Parietal Lobe", "Occipital Lobe", "Insula", "Hypothalamus"}. For chain-of-thought (CoT) prompting strategies, the models additionally provide structured reasoning steps and source citations formatted within the same dictionary structure.

### Large Language Models and Prompt Strategies

In our study, we evaluate a diverse set of large language models (LLMs), both open and proprietary models, on their effectiveness in seizure semiology localization tasks. The models are chosen to represent a range of architectures, parameter sizes, and training methodologies, allowing us to assess the state-of-the-art advancements in natural language processing for specialized medical tasks. **Mixtral-8 × 7B**, developed by Mistral AI (2023)[25], is a sparse mixture-of-experts (MoE) model. This 46.7B parameters decoder-only model employs a unique architecture: each feedforward block selects from 8 distinct expert groups, but only 12.9B parameters are used per token. A router network dynamically selects two experts per token at each layer, combining their outputs additively. As a result, Mixtral performs with the speed and cost efficiency of a 12.9B model while retaining the flexibility of a larger model. The model is pre-trained on open web data, simultaneously

optimizing both experts and router networks. **Qwen-72B**[26] proposed by Alibaba Cloud, comprises 72 billion parameters and follows a transformer-based, decoder-only architecture. We specifically utilize Qwen-72B-Chat model, renowned for its stable 32,000-token context capacity, allowing comprehensive processing of extensive textual inputs. We also incorporated **LlaMA** models developed by Meta, which have gained widespread recognition. Specifically, we test the LlaMA-70B-chat versions from both v2(released in mid 2023) and LlaMA v3 (released in 2024). Additionally, we include proprietary models from OpenAI's **GPT** series. Specifically, we use the gpt-3.5-turbo-1106 model, which represents an advanced iteration of the GPT-3.5 lineage, known for its improved responsiveness and performance over previous versions. Furthermore, we assess gpt-4-1106-preview, the latest available model from OpenAI at the time of experimentation, known for superior comprehension, nuanced reasoning, and human-like text generation capabilities. For all the models we use a temperature of 0.2.

Recently introduced medical models fine-tuned on LlaMA-3 70B version including **OpenBioLLM-70B**[31] (developed by Saama AI Labs) and **Med42-70B**[29] (developed by M42 Health AI Team) are also included in our experiments. These models showcase enhanced capabilities compared to base LlaMA models, however certain models did not follow specialized task instructions, exhibited a tendency to repeat themselves[32], sensitivity to minor changes in the prompt (such as spacing and punctuation) thus lacking flexibility, or restricted context length to accommodate comprehensive symptom descriptions in the prompt[33].

Each LLM is queried with the same symptoms under five prompt strategies to quantify the impact of prompt engineering: Zero-Shot (ZS), Few-Shot (FS), ZS-Chain-of-Thought (CoT), FS-CoT and Self Consistency (SC). **Zero-Shot** prompting uses a structured prompt design for the aforementioned complex task, where the LLM is expected to perform a task based solely on its pre-existing knowledge without any task-specific examples or additional training. **Few-shot** condition follows the in-context learning approach described by Brown et al.[23] and Dong et al.[34], by augmenting the input with K=5 examples to demonstrate the expected input-output structure. This provides the model with representative cases without requiring fine-tuning or retraining. We make use of the term few-shot (FS) and in-context learning (ICL) interchangeably in the manuscript. **Chain-of-Thought (CoT)** method encourages the model to think *step-by-step* and provide intermediate reasoning and sources used to get to the final answer. This technique mimics a human cognitive process, to break complex problems into small, manageable steps. It is helpful where a straightforward answer may not be trivial[35]. To do this, we add a sentence *"Solve the problem in step by step manner"* in the instruction along with the keys "Reasoning" and "Sources". **Few-shot CoT** combines these ideas by providing exemplars from an epileptologist that demonstrate how an expert reasons through the task, with the model expected to learn this pattern in-context and mimic both the reasoning process and output format. Finally, in **Self Consistency (SC)**.[36] we generate multiple independent response chains for each query and select the most consistent response as the final answer. This is done via a majority voting technique. By cross-referencing different outputs, we can ensure that the final response is robust and dependable. In our problem statement, we get likelihoods of 7 brain regions from five reasoning chains. We use median-based majority voting to get the most consistent output. For each brain region $r_i$, compute the median of its likelihoods across five iterations $L_{ij}$ for $j \in \{1, 2, 3, 4, 5\}$, then calculate adjusted likelihoods $A_{ij}$ as the absolute difference from this median:

$$A_{ij} = |L_{ij} - \text{median}(L_{i1}, L_{i2}, L_{i4}, L_{i5})| \tag{2}$$

We then identify the winning iteration $W$ as the one with the minimum sum of adjusted likelihoods across all seven brain regions:

$$W = \arg\min_j \sum_{i=1}^{7} A_{ij} \tag{3}$$

The equations (2) and (3) capture the entire process of majority voting using median activation. The rationale behind using the median is its robustness

to outliers and its representation of the central tendency of the data. In contrast, the mean can be skewed by anomalous data, while the mode, representing the most frequent value, might fail to yield a clear result if all outputs differ.

## Evaluation - Correctness, Confidence and Calibration

Predictions generated by LLMs are determined by selecting the brain region with the highest likelihood value *(argmax)*. These predicted regions are compared against ground truth labels to calculate precision, recall, and F1 scores for each brain region. To account for class imbalance, we use weighted averages of these metrics, with weights proportional to the number of instances in each class. The likelihood output from LLMs is more informative than a single class prediction, as it allows understanding of which classes the model considers plausible, and to what degree, rather than just which class it considers the "winner". We leverage this feature to approximate a confidence/uncertainty measure using Shannon entropy, which is a fundamental concept in information theory that quantifies the "fuzziness" or uncertainty of a system's state. Given a discrete random variable $X$ with possible outcomes $x_1, x_2, ..., x_m$, each with probability $P(x_i)$, the Shannon entropy $H(X)$ is defined as:

$$H(X) = -\sum_{i=1}^{n} P(x_i) \log_2 P(x_i) \tag{4}$$

The entropy ranges from 0 (100% likelihood assigned to one brain region) to 2.807 (likelihoods uniformly distributed across all seven brain regions). We report the loss entropy, defined as (1-normalized entropy), where values approaching 1 mean high model confidence or less uncertain and vice versa if the values tend towards 0.

Furthermore, we evaluate model calibration[37] - alignment between model's predicted probabilities and empirical accuracy - using reliability diagrams and Brier scores. In this work, for each semiology $x_i$, the LLM outputs a likelihood estimate of SOZ of interest. This likelihood estimate of the true class can be represented as $p(y_i|x_i)$. To evaluate calibration across a finite set of semiologies (samples), we partition predictions into $M$ equal width bins of size $1/M$. We then compute the fraction of correct predictions within each bin, commonly known as the empirical accuracy or fraction of positives. Let $B_m$ denote the set of indices of samples whose prediction likelihood falls into the interval $I_m = (\frac{m-1}{M}, \frac{m}{M}]$ where $m \in 1, ..., M$. Then the fraction of positives for bin $B_m$ is defined as:

$$F_{pos}(B_m) = \frac{1}{|B_m|} \sum_{i \in B_m} 1(\hat{y}_i = y_i) \tag{5}$$

where $\hat{y}_i$ and $y_i$ are the predicted and true class labels for sample i, respectively. The average confidence within a bin $B_m$ is calculated as:

$$Conf(B_m) = \frac{1}{|B_m|} \sum_{i \in B_m} p(y_i|x_i) \tag{6}$$

The resulting reliability diagram as shown in Supplementary Fig. S3 provides a visual assessment of model calibration, where perfect calibration is represented by points lying on the diagonal line $F_{pos}(B_m) = Conf(B_m)$. We quantify the calibration using Brier score[38], which measures the mean squared difference between predicted probabilities and actual outcomes. For N samples, the brier score $B_s$ is:

$$B_s = \frac{1}{N} \sum_{j=1}^{N} (p(y_j|x_j) - y_j)^2 \tag{7}$$

## Output Generation and Parsing

To ensure standardized and structured outputs, we explicitly instruct LLMs to generate responses in a predefined JSON format, detailing likelihood percentages for seizure onset zones (SOZ) across brain regions (e.g., "Temporal Lobe": a%, "Frontal Lobe": b%). A custom regex-based parser

validates format integrity and extracts key information, including SOZ likelihood estimates, reasoning chains, and literature citations. This structured approach minimizes variability in the format and improves consistency across responses. In case models returned missing, non-numeric, or ambiguous values (such as "None"), a likelihood of 0% is assigned for the respective brain region.

For our multilingual evaluation, we systematically translate seizure descriptions and brain region names into French, Spanish, and Chinese using DeepL tool, ensuring terminological consistency. Additionally, language-specific parsers are designed to handle these translated terms, enabling reliable structured evaluation across multiple languages while mitigating potential discrepancies arising from variations in terminology or syntax.

## Baselines

As there is no established benchmark for this task, we define a lower bound performance (38.21%) by implementing a naive classifier that always predicts the most probable class in the data. In addition, to assess statistically whether the models produce any meaningful outputs, we conduct a permutation-based significance test. Specifically, we create a random performance distribution by repeatedly (999 times) permuting true labels and recalculating the F1 score of each model. This procedure provides a by-chance distribution of F1 scores, capturing the expected performance if the models' outputs were random. To quantify the deviation of our model's actual F1 score from the random distribution, we perform a Z-score normalization as follows:

$$z = \frac{(x - \mu)}{\sigma} \tag{8}$$

Here, x is the actual F1 score, $\mu$ is the mean F1 score obtained from the random distribution and $\sigma$ is the standard deviation of this random distribution. A high Z-score indicates significantly better performance compared to random chance. The corresponding *p*-values are computed to determine the statistical significance of this deviation. *p*-value < 0.05 indicate that the actual F1 score of the LLM is unlikely to have occurred by chance, providing evidence of the model's effectiveness beyond random performance as shown in Supplementary Table S10.

## User Study

We incorporated expert clinical assessment into our benchmarking framework through two structured online surveys designed to evaluate both predictive performance and reasoning quality. In the first survey, we present 81 semiologies chosen from diverse semiological categories to a clinical expert (author SL, FP). Each participant assigned likelihood values (0-100%) for seizure onset zone (SOZ) localization across seven distinct brain regions using a slider interface as shown in Supplementary Fig. S2a. This design intentionally mirrored the output format of our computational models, enabling direct comparison between clinician and model predictions. Both of the clinicians are neurologist in Germany with a specific focus and training in epilepsy care, including >1 year experience in exclusively treating patients in an Epilepsy Monitoring Unit (SL: equivalent in level to a board-certified neurologist and FP: a board-certified neurologist).

In the second survey [Refer Supplementary Fig. S2(b)], we assess the quality of explanatory content generated by the best-performing models (GPT-4 and Mixtral-8 × 7B). To prevent bias, model identifiers were omitted from the presented reasoning. Clinical expert evaluate output for correctness and completeness across three levels. This enables us to effectively quantify the utility of long-form reasoning generated by models in epilepsy diagnosis. Additionally, two authors of the paper independently verified the accuracy of sources cited within model outputs, confirming both author lists and publication titles verbatim to ensure citation integrity.

## Statistics and Reproducibility

Seizure onset zone (SOZ) localization performance was quantified using F1 scores for each model and prompt strategy. For every value reported in the main F1 score table, we generated 999 bootstrap resamples to estimate the mean and 95% confidence interval and compared these estimates against a random baseline classifier matched to the empirical class distribution. Statistical significance of F1 scores relative to random performance was assessed using two-sided permutation tests, yielding mean, standard deviation, *Z*-scores, and Benjamini–Hochberg-corrected *p*-values for each model-prompt combination (significance threshold $p < 0.05$ for all tests). For the clinical reasoning evaluation, pairwise differences in correctness and completeness proportions between models were analyzed using two-sided *z*-tests for proportions. Comparisons between LLMs and classical machine learning baselines (Supplementary Table S11 and Supplementary Fig. S5) were performed using Wilcoxon signed-rank tests over repeated splits, again applying a significance threshold of $p < 0.05$ in all cases.

## Ethics Statement

This study was performed in accordance with the Declaration of Helsinki[39] and Ethical approval was obtained from the Ethics Committee of the University of Tübingen Medical school (reference no. Ethics-2026-0280-A). Informed consent was waived due to the usage of fully anonymized non-identifiable publicly available data.

## Results

Our experimental pipeline of *semioLLM* is illustrated in Fig. 1. Evaluated LLMs are given a text describing seizure symptoms taken from a public database [22]. In the database, each seizure description is linked to one of 7 brain regions containing the respective seizure onset zone (SOZ). This ground truth was determined based on the fact that post-surgical resection of the SOZ patients remained seizure free for at least one year.

### Prompt strategies significantly boost performance

We assess classification performance for SOZ localization using the F1 score, comparing LLMs against a clinical evaluation and a naive classifier (lower bound, see Methods section). We further compare the zero shot setting (ZS) to 4 different prompt strategies: In the few-shot (FS)[23] condition, we leverage in-context learning (ICL) and provide each model with representative input-output pairs of seizure descriptions and their SOZs. We employ Chain-of-thought (CoT) prompting[35], which provides step-by-step reasoning to improve the models' ability to handle complex reasoning tasks. We also implement a task-specific strategy, FS-CoT, which combines the desired input-output mapping with a reasoning patterns curated by an expert clinician, mimicking the diagnostic reasoning of an epileptologist. Finally, we employ self-consistency (SC)[36], generating multiple reasoning paths and determining predictions through majority voting, which have been shown to enhance inference robustness and produce more reliable predictions for complex tasks.

As shown in Fig. 2(a) top, most models perform just above chance level in the zero-shot (ZS) condition. Exceptions are Mixtral-8 × 7B and GPT-4, which achieved substantially higher F1 scores than all other models of 51.66% and 52.27%, respectively (95% CI: [51.43, 51.90], [52.04, 52.50]), comparable to the clinician's performance: Clinician 1 - 48.77% (95% CI: [48.53, 49.02]) and Clinician 2 - 46.75% (95% CI: [46.51, 46.99]). Importantly, we observed a substantial performance increase when introducing prompt-engineering across all models: Median F1 improvement relative to ZS was 6.49% for Few-Shot prompting, 9.62% for CoT, 9.49% in FS-CoT, and 10.02% increase for SC. Note that GPT-4, maintained a consistently high performance in all conditions, only showing a modest gain from 52.27% in ZS to 53.44% with SC. In addition to general-purpose LLMs, we evaluated two medical-specific models-OpenBioLLM-70B and Med42-70B. OpenBioLLM-70B performed well, particularly in CoT (52.19) and SC (53.06), approaching score in SC (53.44), but showed weaker ZS performance (38.56). Med42-70B exhibited better calibration than OpenBioLLM-70B (refer Fig. 2(b) top), yet lower overall scores (e.g., 46.70 in CoT, 47.63 in

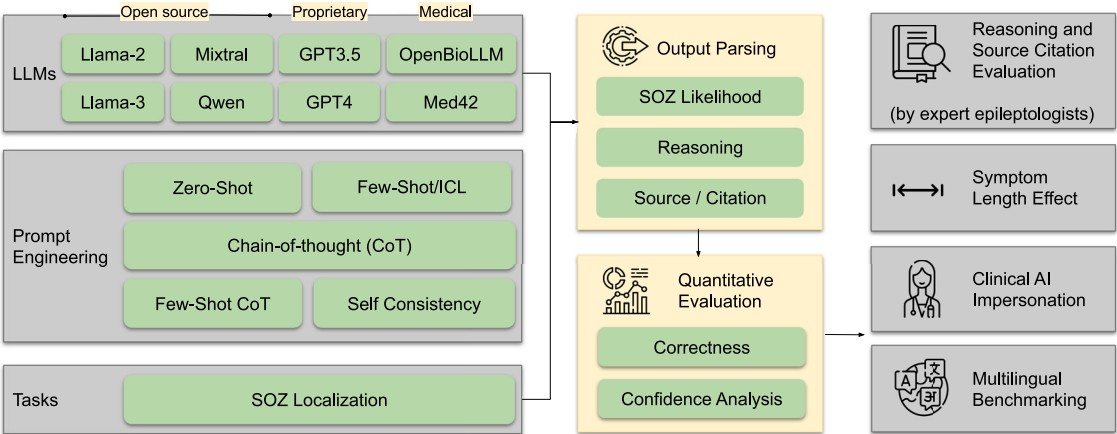

**Fig. 1 | Overview of *SemioLLM*.** We consider eight open-source and proprietary LLMs and evaluate them across five standard prompt styles for the task of SOZ localization. Model outputs include likelihood estimates of seven major brain regions, reasoning and source citations, and are evaluated for accuracy and confidence. The best performing models are examined in further depth, covering task comprehension, logical reasoning, knowledge retrieval, clinical safety and source citation verification, as well as the impact of symptom description length, in-context clinical impersonation, and multilingual alignment and understanding. The icons used in the creation of Fig. 1 are sourced from Flaticon.

SC). While, both medical LLMs approached some general models in specific settings, neither consistently matched the performance of the top general-purpose models such as GPT-4.0 (mean 52.10) or Mixtral-8 × 7B (mean 50.58) - which achieved high accuracy for seizure-onset zone prediction, even without any external guidance (ZS). However, other models are able to match this through prompt-engineering. Interestingly, FS-CoT and SC both demonstrate the highest positive impact even though relying on different strategies-with FS-CoT providing expert-curated reasoning patterns that guide model outputs, and SC by enhancing robustness through the aggregation of multiple independent reasoning paths. Refer Supplementary Table S1 for detailed per-model scores across all prompt strategies.

### High confidence does not guarantee correctness

To move towards trustworthy AI in risk-sensitive domains such as medicine, it is crucial to develop systems that are not only correct but also confident and well-calibrated in their outputs[40,41]. Confidence, in practical terms, implies a higher degree of certainty in predictions. To assess this, we computed an entropy-based measure derived directly from model outputs rather than using a subjective self-assessment by LLMs[41–44]. Specifically, we compute normalized Shannon's entropy ($H$)[45] using the likelihood estimates for different SOZs and derive a confidence score ($C = 1 - H$) that ranges from 0 (lowest confidence) to 1 (highest confidence, see Eq. (4) in Methods). Across all models, confidence scores were lowest in the zero-shot condition and improved consistently with prompt engineering (Fig. 2(a) bottom). For example, providing multiple solution examples in the few-shot condition increased the average model confidence by 13.75% compared to zero-shot, while task-specific reasoning crafted by an epileptologist in the FS-CoT condition resulted in a substantial 21% increase in confidence, indicating that domain-specific demonstrations on task reasoning can significantly enhance the certainty of model predictions. Finally, as expected models exhibited the highest overall confidence for self-consistency (SC) with 35.25% improvement as this prompt style is intended to reduce stochastic variability by aggregating predictions across multiple reasoning paths. Assessing model confidence also requires evaluating calibration-that is how well model's predicted probabilities align with its actual correctness. Even highly accurate models can produce probability estimates that do not reliably reflect their true likelihood of being correct[37]. Calibration can be quantified using the Brier score loss[38,46], where lower scores indicate better alignment. As shown in Fig. 2(b) top, models exhibit a high variance in calibration in the zero-shot condition. This markedly improves for all models with more refined prompting techniques. Similar to the effect we observed when assessing model confidence through entropy, FS-CoT and

SC are most efficient for aligning predicted probabilities more closely with actual accuracy. Notably, GPT-4 shows the best calibration, even in the zero-shot condition. Refer Supplementary Table S2 and S3 for detailed per-model scores across all prompt strategies.

Figure 2 (b) bottom provides a summary of model performance across three key dimensions (average across prompt styles): accuracy (F1 score), confidence, and calibration. Larger enclosed areas indicate models that better balance these factors, highlighting trade-offs between predictive accuracy and reliability. Given the medical domain's high-risk nature, the ideal model is one that not only achieves strong performance but also maintains well-calibrated confidence, reducing the likelihood of overly certain but incorrect decisions. Overall, two models - GPT-4 and Mixtral-8 × 7B - strike the best balance between all factors in our task. We therefore consider these two models for a more detailed investigation, including reasoning analysis and experimental manipulations such as in-context impersonation, narrative length effects, and cross-linguistic comparisons.

### Evaluating Clinical Reasoning and Source Attribution

Thus far, our results demonstrate that LLMs can effectively map unstructured seizure descriptions to seizure onset zones (SOZs) in the brain. To explore how LLMs arrive at their decisions, we assessed the reasoning abilities of the two best performing models, GPT-4 and Mixtral-8 × 7B on a randomly selected subset of 81 chain-of-thought (CoT) responses using a clinical evaluation (see Supplementary Fig. S2 and Methods for details). Following Med-PaLM[7] and Liévin et al.'s[47] protocol, we evaluated our models' reasoning outputs using three categories: correct/complete, somewhat correct/complete, and incorrect/incomplete. Additionally, each output is assessed for the proportion of correct and incorrect statements along three dimensions: (i) comprehension, (ii) knowledge recall, and (iii) logical reasoning. A representative example output, where both models correctly identify the seizure onset zone (SOZ), along with the annotations, is shown in Fig. 3(a). For the semiology "right upper limb ballistic-like movement", both models demonstrate accurate comprehension, knowledge recall and logical reasoning (yellow, blue and green). However, in contrast to GPT-4, Mixtral-8 × 7B misinterprets the associated hemisphere leading to an error (see pink text: "right frontal lobe"). Additionally, Mixtral includes incorrect supporting scientific evidence, while GPT-4 cites a well-fitting and existing paper. These differences are well reflected in our summary analysis shown in Fig. 3b-d. As shown in Fig. 3(b) GPT-4 significantly outperformed Mixtral in both correctness (56.79% vs. 29.63%; z-test for proportions, p <0.05) and completeness (65.00% vs. 34.57%; z-test for proportions, p <0.05). Mixtral's outputs were more often rated as "somewhat correct" and "somewhat

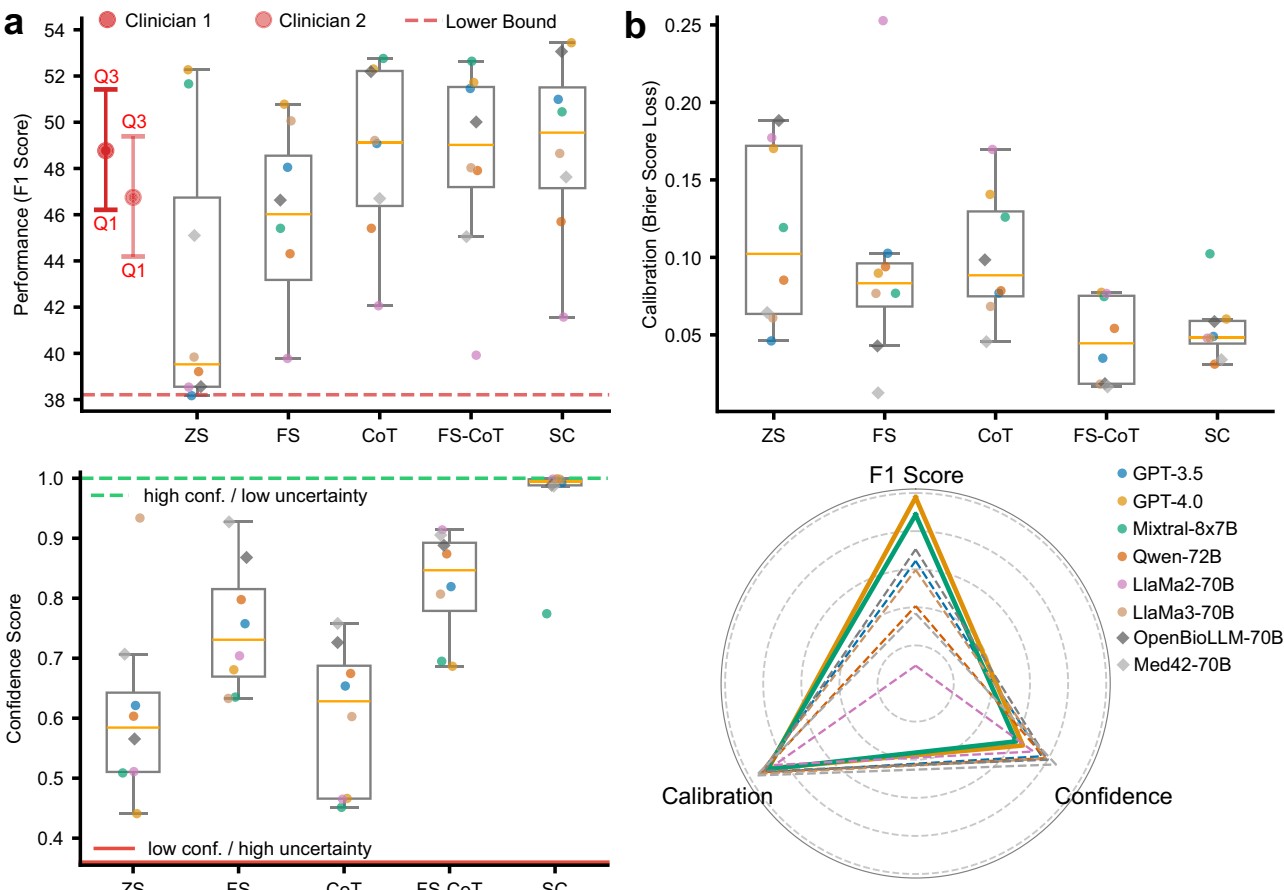

**Fig. 2 | Comparison of LLMs (6 general, 2 medical) and impact of prompt engineering strategies [Zero-Shot (ZS), Few-Shot (FS), ZS-Chain-of-Thought (CoT), FS-CoT and Self Consistency (SC)] on performance, confidence and calibration.** (**a**) Top: Mean F1 scores from 999 bootstrap samples of n=1269 symptoms (n=8 models per prompt style) F1 scores for all models obtained by bootstrapping. Advanced prompt styles achieve performance comparable to clinicians and at par with naive classifier (38.2%,red dashed line) (**a**) Bottom: Mean confidence scores (0=low, 1=high) from 999 bootstrap iterations of n=1269 symptoms (n=8 models per prompt style). In-context learning improves confidence, with FS and FS-CoT showing largest gains (**b**) Top: Calibration via Brier Score Loss from n=1269 symptoms (n=8 models per prompt style); lower indicates better calibration. FS-CoT and SC show best calibration. (**b**) Bottom: Multidimensional performance visualization comparing model correctness, confidence, and calibration metrics, with solid lines representing the best-performing models. Box plots in a-b show the interquartile range (IQR; 25th-75th percentile) with the median indicated by the orange line. Points represent individual models (circles: general LLMs; diamonds: medical LLMs).

complete" ((43.21%, 33.33% respectively). When examining specific dimensions (Fig. 3(c)), GPT-4 made fewer errors for knowledge recall than Mixtral (17.28% vs. 43.21%; z-test for proportions, $p < 0.05$). However, the strongest differences were observed for logical reasoning: GPT-4 maintained a significantly higher score compared to Mixtral (98.77% vs. 80.25%; z-test for proportions, $p < 0.05$). Similarly, Mixtral's incorrect reasoning rate was almost twice that of GPT-4 (38.27% vs. 20.99%; z-test for proportions, $p < 0.05$). Refer Supplementary Table S8 and Supplementary Table S9 for detailed test statistics. Additionally, Fig. 3(d) demonstrates GPT-4's superior citation accuracy (76% for GPT-4 vs. 19% for Mixtral), indicating GPT-4's enhanced ability to provide evidence-based sources for its decision. Refer Supplementary Fig. S4 for comparison between Llama-v2 and v3 models.

### Factors influencing LLM performance in seizure diagnostics
**Symptom description length.** Symptom descriptions vary in length-while longer descriptions may offer more information, they can also introduce irrelevant or contradictory details. To assess how this affects LLM-based diagnostic prediction, we categorized descriptions into six bins - based on word count and calculated the corresponding F1 scores, reporting the mean for each bin across all prompt strategies (refer Fig. 4(a); exact values in Supplementary Table S4)). Interestingly, both models exhibit a distinctive U-shaped pattern, where predictions based

on very short symptom descriptions initially achieve high performance, performance then decreases with increasing description length, and ultimately, observe the highest performance for the longest symptom descriptions. A Kruskal-Wallis test revealed statistically significant differences between the bins ($p<0.001$ for both GPT-4 and Mixtral-8 × 7B). We validated this pattern against surrogate data obtained by randomly sampling semiologies from the original data, preserving the bin sizes. We then recomputed the F1 scores for these shuffled bins. Kolmogorov-Smirnov test ($p<0.001$) between the original and shuffled data reveals that this U-shaped relationship represents a genuine effect of description length rather than a statistical artifact.

**Clinical in-context impersonation.** Clinical decision-making requires context-specific expertise. While prior work has shown that LLMs can improve performance by adopting expert personas in non-medical tasks [48], we investigated whether this extends to clinical diagnostics. We prompted models to adopt increasingly specialized medical roles-medical assistant, neurologist, expert epileptologist-alongside a baseline AI assistant. Each prompt is prefixed with "you are a {persona}" and evaluated in a zero-shot setting to avoid few-shot example confounds. For both GPT-4 and Mixtral-8 × 7B, performance, and confidence improved with increasing domain specificity, with the "expert epileptologist"

**Fig. 3 | Evaluation of model reasoning for representative random sample (n=81 queries) of dataset.** (**a**) Example query and corresponding annotations for a given semiology from GPT-4 and Mixtral-8 × 7B (**b**) Correctness and completeness of model outputs assessed by one clinician (**c**) Breakdown of model performance in reading comprehension, knowledge recall, and reasoning accuracy assessed by one clinician (**d**) Average citation accuracy across models assessed independently by two raters (inter-rater reliability: Cohen's kappa=0.73 for GPT-4, kappa=0.74 for Mixtral-8 × 7B). Bars represent proportion of correct responses out of 81 queries.

yielding the highest scores (Fig. 4(b); exact values in Supplementary Table S5 and S6). GPT-4 showed a substantial performance gain of 13.68%, while Mixtral achieved a smaller but notable improvement (4.47%). Confidence scores also increased with more specialized personas (GPT-4: 9% and Mixtral: 8%). These results demonstrate that in-context persona adaptation enhances both performance and confidence in clinical tasks, with GPT-4 more effectively leveraging contextual cues.

**Multilingual Performance**. To assess the utility of LLMs for cross-lingual clinician-patient communication-particularly in medical tasks that depend entirely on verbal symptom descriptions-we compared performance across three language settings. In the baseline setting, both symptom descriptions and reasoning prompts were in English. In the mixed-language setting, symptom descriptions were in French, Spanish, or Chinese, while prompts remained in English. In the fully translated setting, both the clinical information and the reasoning instructions were presented entirely in French, Spanish, or Chinese, requiring full cross-lingual comprehension and reasoning. Our results (Fig. 4(c); data in Supplementary Table S7) show that both models perform best when prompt and input are in English (En → En), likely reflecting their English-centric training. Interestingly, in the cross-language setting (non-English symptom, English prompt), performance only drops slightly (GPT-4: -1.91%, Mixtral: -2.41%, n.s.). However, in the same-language setting (non-English symptoms and prompt), Mixtral's performance declines substantially by 8%, while GPT-4 remains stable (-1.4%, n.s.). This suggests both models can incorporate non-English

input when anchored with English prompts, but especially Mixtral struggles in non-English contexts.

## Discussion

Previous applications of LLMs in epilepsy have assessed general medical knowledge through structured Q&A formats using single models such as ChatGPT[49–52], while NLP approaches in epilepsy have used rule-based or supervised models trained for narrow tasks like seizure type classification and frequency extraction[53–55]. Most studies evaluated factual recall, rather than diagnostic reasoning and lacked grounding in real-world patient data. In contrast, our work *SemioLLM* presents the first large-scale evaluation of 8 LLMs in diagnostic reasoning from over 1200 unstructured seizure descriptions. We observe that most LLMs can probabilistically infer seizure onset zones without structured input or domain-specific fine-tuning significantly above chance. Notably, GPT-4 and Mixtral-8 × 7B achieve performance comparable to a manual clinician-based assessment even under zero-shot conditions. Prompt engineering led to significant improvements in accuracy, confidence and calibration across all models - specifically clinician-guided chain-of-thought prompting led to the most substantial and consistent improvements. Our findings thus extend previous studies demonstrating the effectiveness of prompt engineering for structured medical Q&A[7], clinical name entity recognition[56], and medical text summarization[57] to probabilistic reasoning from free-text symptom descriptions and especially shows the potential for integrating clinician expertise into foundation model-based clinical decision systems[58–60].

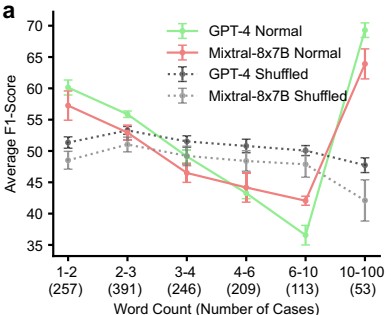
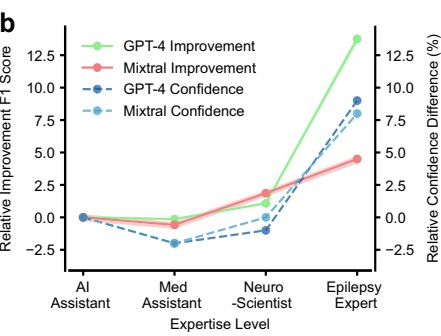
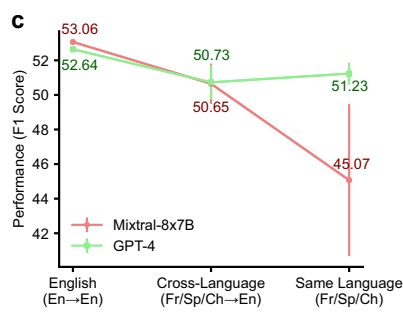

**Fig. 4 | Impact of description length, persona adaptation and language on model performance.** (**a**) Performance across various description-length bins and length-shuffled inputs. Data show means ± SEM. Note that each range [x, y) includes x but excludes y (**b**) Influence of in-context persona adaptation on zero-shot performance, shown as changes in F1 score (red/green) and confidence (blue/dark blue) relative to the AI assistant persona (n = 1269 semiologies). (**c**) Effect of language variation on performance. In the "English" condition, both the semiology description and prompt were in English. In the "Cross-Language" condition, only the semiology description was in a different language. In the "Same Language" condition, both the prompt and semiology description were in a non-English language. Data show means ± SD (n = 1269 semiologies).

A key strength of LLMs is their capacity to generate explanatory reasoning alongside predictions-an important feature for transparency, interpretability, and trust in clinical decision support[61]. Adapting established evaluation protocols from Med-PaLM[7] and Liévin et al.[47], we evaluated reasoning quality along with text comprehension, knowledge recall, and logical inference. Despite similar quantitative prediction performance, GPT-4's reasoning outputs were more often rated as correct and complete in contrast to Mixtral, with a particularly strong advantage in logical inference. In contrast, Mixtral made reasoning errors in over a third of its outputs. Differences in citation accuracy were particularly strong (GPT-4: 76%, Mixtral: 19%), reflecting ongoing challenges in factual grounding and source attribution for generative models[62,63]. These results underscore that performance metrics alone may obscure reasoning deficiencies, findings that are known from studies testing general medical knowledge [8]. Here, retrieval-augmented generation (RAG)[64,65] may help ground LLM reasoning in accurate, up-to-date knowledge, possibly improving reliability without retraining.

Our study also reveals critical factors influencing LLM performance on this task. First, similar to prior results in non-medical domains[66], emulating an increasingly aligned clinical expert systematically improved both performance and confidence by 14% and 10% across models, respectively. Context-specific impersonation thus augments the ability of generalized language models to perform domain specific clinical tasks. Second, prediction performance varied with symptom description length, showing accuracy differences of up to 32%, where very short and highly detailed descriptions outperformed intermediate-length narratives. A possible explanation might be that brief descriptions closely match distinct canonical seizure features (e.g., "visual aura" for occipital onset[67]), resembling the benefits of concise N-gram-based inputs in NLP tasks[68]. Increasing description length, however, may introduce redundancies or contradicting evidence potentially degrading model performance [69]. However, richly detailed but coherent input may offer enough structured context for LLMs to disambiguate and reason over complex clinical information. Ultimately these findings implicate that the level of detail as approximated through symptom description length might similarly influence diagnostic accuracy in LLMs and clinicians[70]. Third, our evaluation showed that current LLMs can generalize across languages when anchored with English prompts, but also revealed limitations when all inputs are non-English. Both top-performing models, GPT-4 and Mixtral showed robust multilingual generalization when reasoning prompts were presented in English, even when symptom descriptions were provided in other languages such as French or Chinese, suggesting that they can integrate multilingual input when anchored with English-language instructions. In contrast, performance for one model (Mixtral) declined substantially when both prompts and input symptoms were non-English. This likely reflects the English-dominance of

instruction tuning and pretraining corpora and points to limited cross-lingual generalization in current LLMs for the clinical domain, similar to limited multilingual performance in general-purpose NLP tasks [71]. Targeted multilingual instruction tuning, particularly in clinical domains, may thus be necessary not only to ensure robust and inclusive model behavior but also equitable application in multilingual healthcare systems.

Unlike prior approaches that assess factual recall or classification, our framework combines quantitative metrics-correctness, confidence, and calibration - with qualitative expert annotation and evaluation of model reasoning. Importantly, LLMs can transform free-text clinical narratives into structured, actionable diagnostic inferences. Our work thus moves beyond basic knowledge verification and toward real-world clinical applicability. Importantly, our framework can be adapted to other medical domains. As a proof of concept, we provide example code and preliminary analyses in our GitHub repository in the domain of dermatology, linking skin anomaly descriptions to diagnostic classes.

Despite these strengths, our study has several limitations. Our analysis is restricted to a single dataset which includes only adult focal epilepsy cases. While the diagnostic task (localization of SOZ) may apply to pediatric patients, it is not suitable for generic, generalized seizures. Additionally, while the dataset includes cases from diverse sources, we did not have the necessary metadata to analyze the impact of demographic and cultural variations in patient descriptions. While our translation-based cross-lingual analysis suggests that the language of seizure descriptions can affect performance, a key limitation is that our original corpus was monolingual. It will therefore be important in future studies to quantify language-specific effects in culturally diverse seizure descriptions. We demonstrate in our public code repository that the framework can be adapted to other clinical domains such as dermatology, our systematic evaluation and prompt optimization were conducted specifically for epilepsy. Consequently, prompt engineering strategies may need to be re-evaluated and tailored for other specialties before clinical deployment. Finally, as a comprehensive manual annotation would require impractical amounts of clinician effort, our reasoning analysis focused on a representative subset, chosen to match the overall distribution of the data. While this approach provides practical insight, it may introduce a degree of sampling bias or subjectivity, and future studies with larger-scale or multi-annotator reasoning evaluations would further strengthen the conclusions.

## Data availability

All data are made available within the article, supplementary information, or the source data file provided with this paper. The Semio2Brain dataset[22] used in this study is publicly available at https://doi.org/10.5281/zenodo. 4606589[72]. Source data for Figs. 2(a,b), 3(b,c,d), and 4(a,b,c) are provided as individual sheets within a single Supplementary Data file accompanying

this paper. All source code required to reproduce the analyses and figures is publicly available via GitHub repository: https://github.com/liebelab/semiollm and archived on Zenodo[73]: https://doi.org/10.5281/zenodo.18455275.

## Code availability

The source code to reproduce the findings is present on public Github Repository https://github.com/liebelab/semiollm[73]. Persistent link to the version of the code described in this paper are available at Zenodo[73]: https://doi.org/10.5281/zenodo.18455275.

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

## Acknowledgements

This work was supported by the Else Kröner Fresenius Foundation, German Research Foundation (DFG, 493665037), SPP 2241 - PN 520287829, the Machine Learning Cluster of Excellence EXC number 2064/1 PN 390727645, Tübingen AI Center, SFB1233 'Robust Vision', ERC (853489-DEXIM). BMFTR (01GQ2502), BMBF (01IS18039A), and the Alfried Krupp on Bohlen und Halback Foundation. M.D. is a member of the International Max Planck Research School for Intelligent Systems Tübingen (IMPRS-IS). The authors thank Prof. Dr. Jakob H. Macke and his lab for discussion and feedback; Almut Sophia Koepke, Shyamgopal Karthik, Matthias Tangemann and Elisa Nguyen for comments on the manuscript.

## Author contributions

M.D.: Conceptualization, Methodology, Validation, Formal Analysis, Data Curation, Reasoning Study Design and Analysis, Visualization, Writing - Original Draft, Writing - Review and Editing. M.J.P.: Methodology (design of clinical study form); Formal Analysis (compilation and evaluation of clinical responses and model source citations), Writing - Review and Editing. F.P.: Clinical evaluation, Writing - Review. Z.A.: Supervision, Funding Acquisition. S.L.: Conceptualization, Methodology, Clinical evaluation, Project Administration, Supervision, Funding Acquisition, Writing - Review and Editing.

## Funding

## Competing interests

The authors declare no competing interests
