## [Transparent Peer Review file · Communications Medicine]

Evaluating Large Language Models for Diagnostic Reasoning from Unstructured Clinical Narratives in Epilepsy

Corresponding Author: Ms Meghal Dani

Version 0:

Reviewer comments:

Reviewer #1

(Remarks to the Author)

This study presents SemioLLM, a framework evaluating six state-of-the-art LLMs (GPT-3.5, GPT-4, Mixtral-8x7B, Qwen-72B, LLaMa2, LLaMa3) in predicting seizure onset zones from 1,269 unstructured clinical descriptions. The findings demonstrate that LLMs, particularly with chain-of-thought prompting and clinical impersonation, can approach clinician-level accuracy (improving performance by 10% and 13.8%, respectively). However, performance varies with narrative length (showing a U-shaped relationship) and language context, with English-trained models struggling in non-English prompts. While GPT-4 and Mixtral-8x7B balance accuracy and confidence well, expert analysis reveals concerning reliance on hallucinated reasoning and poor source citation, limiting clinical interpretability. The study provides valuable insights into LLMs' diagnostic potential but underscores the need for improved reliability and multilingual adaptation before real-world deployment.

While this study provides valuable insights into LLMs' diagnostic capabilities, several limitations should be noted. First, the evaluation relies on a single annotated dataset, which may not fully represent the diversity of real-world clinical narratives. Second, the assessment of model reasoning and hallucinations is based on expert analysis of a subset of cases, potentially introducing subjectivity. Third, the study focuses primarily on epilepsy, limiting generalizability to other medical domains. Additionally, the impact of demographic and cultural variations in patient descriptions is not explored. Finally, while prompt engineering improves performance, the optimal strategies may vary across healthcare systems and languages, requiring further validation.

Reviewer #2

(Remarks to the Author)

This is a very well done study overall.

MAJOR

- In an effort to be intellectually honest, the abstract and discussion should be more explicit about what this system is capable of, and isn't. Given a brief phrase of semiology alone (along with some special filtering to make the text better) coupled with the strategies the authors used, results that appear to often match the ground truth came back. That is different from saying you just put in any free text and out comes the localization. As long as this seemingly minor point is made clear, the rest is great.

MINOR

- Remove "state of the art SOTA" from descriptions of outdated versions of LLMs given that all of the models tests are no longer current.

Reviewer #3

(Remarks to the Author)

Brief summary of the manuscript

This study evaluates six popular large language models (Mixtral, Qwen, 2xLlama, and 2xGPT) in predicting the seizure onset zones based on free-text descriptions of seizure semiology in epilepsy patients. The study utilized a publicly available seizure semiology dataset from ~1,200 patients. The LLMs were prompted using zero-shot, few-shot, chain-of-thought and self-consistency prompting strategies and were evaluated by clinicians for correctness and reasoning capability.

Results show that GPT-4 and Mixtral-8x7B achieve performance comparable to clinicians even under zero-shot conditions and those performances further improved with prompt engineering. However, the GPT-4 performed significantly better than Mixtral in reasoning capability. Study also performed evaluations using prompts in languages other than English which showed some robustness for GPT in multi-lingual settings.

Overall impression of the work

The paper is well-written and highlights the value of LLMs in challenging clinical decisions such as the determination of seizure onset zones, while also discussing their limitations. The evaluations were performed using a large dataset which adds value to the findings. However, I have several major concerns regarding the experiments and the main scientific contributions of this paper.

Specific comments, with recommendations for addressing each comment

1. The goal of this paper is unclear. Is it that the authors are trying to validate an LLM-based tool for a particular medical task or is it that they are trying to comprehensively assess LLMs and make general conclusions about their use in epilepsy? In either scenario, the experiments are not substantial enough for the claims made. If it is the former, we would need more extensive baselines and if it is the latter, we would need more relevant LLMs, potentially ones that are medical-specific. See below comments.
2. A main concern I have is that clinician performance is measured from only one expert. This limits the validity of the main claim of the paper that LLMs perform similar or better than clinical experts.
3. The two baselines considered are not fair baselines. As far as I understand, the dataset is categorical and therefore I believe that many machine learning models can be trained using this data. It will be useful to understand how classical ML models perform on this task.
4. The LLMs considered in this work are general purpose LLMs and are not specific to medical text. There are many medical-specific LLMs that are more suitable for this task, Med-Llama, Med42, Med-PaLM, Phi-4, etc.
5. Broadly, this paper appears to be simply an evaluation of general LLMs on a publicly available data and does not generate any data or clinical knowledge. For example, if the authors could explain why in some cases the LLMs are better than clinicians and evaluate whether those explanations can improve the current clinical knowledge, that would significantly enhance the value of this paper.

Minor comments:

1. Figure 3 caption: there are two (b)'s.
2. In figure 4c caption, there is a discrepancy between the x-axis tick label 'English' and the caption saying 'Same English'.

Version 1:

Reviewer comments:

Reviewer #1

(Remarks to the Author)

The manuscript has improved significantly and now presents a thoughtful, rigorous evaluation of LLMs for semiology-to-SOZ mapping. Thank you for the detailed rebuttal and the substantial revisions made. The authors have clearly invested considerable effort in addressing the reviewers' concerns, and the updates are well executed. I have no further comments.

Reviewer #2

(Remarks to the Author)

Overall much improved.

Minor:

- please specify what is the training qualifications of "clinician 1" and "clinician 2" since a significant claim is that the AI tool is on par with them
- Please double check if 2 sig. figs after the decimal are really needed for percentages quoted.

Reviewer #3

(Remarks to the Author)

I thank the authors for the extensive work they did to address my comments. I have no further comments.

Rebuttal

SemioLLM: Evaluating Large Language Models for Diagnostic Reasoning from Unstructured Clinical Narratives in Epilepsy

Meghal Dani^{1,2} Muthu Jeyanthi Prakash¹ Filip Rosa³ Zeynep Akata^{2,4} Stefanie Liebe^{3,5}

We sincerely thank all three reviewers for their insightful and positive feedback on our manuscript. We have made substantial revisions and included new analyses based on their suggestions and have updated the manuscript accordingly. We believe that our updates have significantly strengthened the manuscript. While we discuss each point raised in detail below, we have made the following main additions:

- We now include two open-source medical LLMs in our evaluation analysis - Med42-70B [3] and OpenBioLLM-70B¹. In brief, one of the medical models and one ML algorithm performed overall similar to general LLMs.
- We included a second clinical expert to independently evaluate all seizure descriptions. Performance was similar between both evaluators, and did not change our overall findings.
- We demonstrate the adaptability of our approach across clinical contexts using dermatology as a preliminary example. We include this analysis only in the rebuttal and our Github repository, as it extends beyond the paper's scope.
- We have updated the abstract and introduction to more clearly describe the task, inputs, and scope of our approach. We have revised the discussion to note current limitations - including the focus on adult epilepsy and lack of investigating socio-demographic and cultural variables.

R1: Review *This study presents SemioLLM, a framework evaluating six state-of-the-art LLMs (GPT-3.5, GPT-4, Mixtral-8x7B, Qwen-72B, LLaMA-2, LLaMA-3) in predicting seizure onset zones from 1,269 unstructured clinical descriptions. The findings demonstrate that LLMs, particularly with chain-of-thought prompting and clinical impersonation, can approach clinician-level accuracy (improving performance by 10% and 13.8%, respectively). However, performance varies with narrative length (showing a U-shaped relationship) and language context, with English-trained models struggling in non-English prompts. While GPT-4 and Mixtral-8x7B balance accuracy and confidence well, expert analysis reveals concerning reliance on hallucinated reasoning and poor source citation, limiting clinical interpretability. The study provides valuable insights into LLMs' diagnostic potential but underscores the need for improved reliability and multilingual adaptation before real-world deployment. While this study provides valuable insights into LLMs' diagnostic capabilities, several limitations should be noted.* **Response to R1:**

- *First, the evaluation relies on a single annotated dataset, which may not fully represent the diversity of real-world clinical narratives.*

We acknowledge the limitation raised by the reviewer on using one annotated dataset only, which may limit generalizability. We have updated the discussion accordingly (page 8, lines 395–399).

Nevertheless, to the best of our knowledge, the Semio2Brain dataset [1] is currently the most comprehensive publicly available resource in focal epilepsy symptoms. In general, other published datasets for language analysis in epilepsy are substantially smaller, such as the recent study by Fonferko et al. [5] and Mora et al. [10], which used 200 annotated clinical notes and 536 seizure descriptions (not publicly available and only comprising temporal and frontal lobe epilepsy), respectively. With over 1,200 seizure descriptions, our dataset is therefore larger than many domain-specific

¹University of Tübingen, Tübingen, Germany ²Helmholtz Munich, Munich, Germany ³Dept. of Neurology and Epileptology, University Clinic Tübingen, Hertie Institute for Clinical Brain Research, Tübingen, Germany ⁴Technical University of Munich, Munich, Germany ⁵Machine Learning in Science, Excellence Cluster Machine Learning, Tübingen University, Germany. Correspondence to: Meghal Dani <meghal.dani@uni-tuebingen.de>, Stefanie Liebe <stefanie.liebe@uni-tuebingen.de>.

Preliminary work under review. Do not distribute, Copyright 2025 by the author(s).

¹<https://huggingface.co/aaditya/Llama3-OpenBioLLM-70B>

corpora. Large-scale general medical benchmarks (e.g., MedMCQA [12]) may include tens of thousands of general medicine examples but they are not domain-specific.

- *Second, the assessment of model reasoning and hallucinations is based on expert analysis of a subset of cases, potentially introducing subjectivity.*

We would like to clarify that our analysis employs random sampling to reflect the underlying distribution of semiological categories in the full dataset. Our revised sample now includes 81 (formerly 58) semiologies. Notably, as shown in the updated **Figure 3** in the main paper and **Figure 1** here, the results remain consistent with our original findings: for GPT-4, the proportion of responses rated as “correct” and “complete” remained high (correctness: 55.17% to 56.79%, completeness: 65.51% to 65.00%), while Mixtral’s responses continued to be more frequently rated as “somewhat correct” (44.82% to 43.21%) and “somewhat complete” (32.75% to 33.33%). Other dimensions—including knowledge recall, logical reasoning, and citation accuracy—showed **similar trends** to the original results, confirming the robustness of our findings.

- *Third, the study focuses primarily on epilepsy, limiting generalizability to other medical domains.*

We now show the adaptability of our framework for other medical domains (dermatology, pulmonology). We provide an example preliminary analysis in this rebuttal for dermatology, which required minimal changes—only a few domain-specific variables (diagnostic outcomes, symptom descriptions, domain personas) were substituted in the main prompt. For dermatology, skin anomalies were mapped to classes such as “Acne”, “Fungal infection”, “Impetigo”, “Psoriasis”, and “others”. We used a publicly available dataset on Kaggle Symptom2Disease², which contains labeled symptom collections. Preliminary results for one model (GPT-4.0) as shown in **Figure 3** here in the rebuttal, are consistent with our main findings in epilepsy: chain-of-thought prompting and the use of domain-specific clinical persona (dermatologist in this case) substantially improved accuracy (e.g., accuracy increased from 55% in the zero-shot setting to 70% with CoT and 30% with AI-assistant persona to 55% with dermatologist). While we think an in-depth investigation into other medical domains is out of scope of this study, we have added experimental details in the associated GitHub repository³. The findings are briefly summarized in the Discussion (page 8, lines 390-394, 401-404) to highlight the generalizability of our method.

- *Additionally, the impact of demographic and cultural variations in patient descriptions is not explored. Finally, while prompt engineering improves performance, the optimal strategies may vary across healthcare systems and languages, requiring further validation.* Unfortunately, cultural and demographic information were not available along with seizure descriptions in the current database, which limits us to investigate the impact of these factors on model performance. We have noted this limitation in the discussion section. Regarding prompt strategies, we tested several manipulations. We would argue that some are more generalizable than others, such as providing examples for correct predictions (i.e. few-shot), or logical reasoning on solving the task (few shot chain-of-thought) [2, 7, 13, 17]. With respect to language variations we investigated cross-lingual performance (using French, Chinese, Spanish) and showed indeed a drop in performance for one model (Mixtral) vs. GPT-4.0 (**refer Figure 4c of the main manuscript**). While this does not specifically address cultural - or language specific differences in seizure descriptions, it highlights the importance of multi-lingual training in these models. Nevertheless, we acknowledge the importance of the raised points and comment on this in the discussion section of the manuscript (page 7 and 8, lines 377-387, 395-401)

R2: Review

This is a very well done study overall.

Response to R2:

- *MAJOR: In an effort to be intellectually honest, the abstract and discussion should be more explicit about what this system is capable of, and isn’t. Given a brief phrase of semiology alone (along with some special filtering to make the text better) coupled with the strategies the authors used, results that appear to often match the ground truth came back. That is different from saying you just put in any free text and out comes the localization. As long as this seemingly minor point is made clear, the rest is great.*

²<https://www.kaggle.com/datasets/niyarrbarman/symptom2disease/data>

³<https://github.com/liebelab/semiollm>

a Query Semiology: right upper limb ballistic-like movement

GPT4

Reasoning: The description of ballistic-like movement typically refers to sudden, involuntary flinging motions of the limbs. These movements are often associated with lesions or seizures involving the subthalamic nucleus, but when they are part of a seizure semiology, they are more commonly associated with the frontal lobe, particularly the precentral gyrus, which is involved in motor control. The fact that it is the right upper limb suggests that the seizure onset zone is likely in the left hemisphere, as motor control is typically contralateral. However, without more specific information about the nature of the movements and their context, it is difficult to provide a definitive localization. The likelihood of other regions being involved is low, but not impossible, as complex motor movements can sometimes have a more widespread network involvement.

Source: Tatum, W. O., Rubboli, G., Kaplan, P. W., Mirsafari, S. M., Radhakrishnan, K., Gloss, D. & Koutoumanidis, M. (2018). Clinical utility of EEG in diagnosing and monitoring epilepsy in adults. *Clinical Neurophysiology*, 129(5), 1056-1082

Likelihood: "Temporal Lobe": "0%", "Frontal Lobe": "90%", "Cingulate Gyrus": "0%", "Parietal Lobe": "10%", "Occipital Lobe": "0%", "Insula": "0%", "Hypothalamus": "0%"

Mixtral8x7B

Reasoning: The patient is showing a repetitive, involuntary movement of the right upper limb, which is described as ballistic-like. Ballistic movements are rapid, forceful, and can be large amplitude, and are often seen in hyperkinetic movement disorders. In epilepsy, these movements are indicative of a seizure onset zone in the motor cortex, which is located in the frontal lobe. The lateralization to the right side indicates that the seizure onset zone is likely located in the right frontal lobe

Source: Jehi, L., Pellinen, T., Widjaja, E., & Hallett, M. (2012). Epileptic motor phenomena: clinical and electrophysiological correlations. *Epilepsia*, 53(10), 1681-1691.

Likelihood: "Temporal Lobe": "0%", "Frontal Lobe": "100%", "Cingulate Gyrus": "0%", "Parietal Lobe": "0%", "Occipital Lobe": "0%", "Insula": "0%", "Hypothalamus": "0%"

Correct reasoning step (green), Incorrect reasoning step (pink), Correct recall of knowledge (blue), Incorrect or insufficient knowledge (purple), Correct reading comprehension (yellow), Incorrect reading comprehension (red)

Figure 1. Evaluation of model reasoning for representative random sample of dataset. (a) Example query and corresponding annotations for a given semiology from GPT-4 and Mixtral-8x7B (b) Correctness and completeness of model outputs (c) Breakdown of model performance in reading comprehension, knowledge recall, and reasoning accuracy (d) Comparison of average citation accuracy across models. Note that Inter-rater reliability was high, with Cohen’s kappa scores of 0.70 for GPT-4 and 0.77 for Mixtral-8x7B, indicating strong agreement between evaluators.

We thank the reviewer for their overall positive evaluation of our study. We have updated our abstract to be more clear to the following: “In this study we task eight Large Language models (GPT-3.5, GPT-4.0, Mixtral-8x7B, Qwen-72B, LLaMA-2, LLaMA-3, OpenBioLLM, Med42) with a diagnostic task in epilepsy: mapping seizure description phrases—after targeted filtering and standardization—to one of seven possible seizure onset zones using likelihood estimates. Most models yield results that often match the ground truth and even approach clinician-level performance after prompt engineering.” In the discussion we highlight the capabilities and limitations of the current approach to be more transparent on page 8, lines 395 - 408.

- *MINOR: Remove “state of the art SOTA” from descriptions of outdated versions of LLMs given that all of the models tests are no longer current.*

Thank you for highlighting this. We have removed all references to “state of the art” or “SOTA” from the manuscript as suggested.

R3: Review This study evaluates six popular large language models (Mixtral, Qwen, 2xLlama, and 2xGPT) in predicting the seizure onset zones based on free-text descriptions of seizure semiology in epilepsy patients. The study utilized a publicly available seizure semiology dataset from 1,200 patients. The LLMs were prompted using zero-shot, few-shot, chain-of-thought and self-consistency prompting strategies and were evaluated by clinicians for correctness and reasoning capability. Results show that GPT-4 and Mixtral-8x7B achieve performance comparable to clinicians even under zero-shot conditions and those performances further improved with prompt engineering. However, the GPT-4 performed significantly

165 better than Mixtral in reasoning capability. Study also performed evaluations using prompts in languages other than English
 166 which showed some robustness for GPT in multi-lingual settings. The paper is well-written and highlights the value of LLMs
 167 in challenging clinical decisions such as the determination of seizure onset zones, while also discussing their limitations.
 168 The evaluations were performed using a large dataset which adds value to the findings. However, I have several major
 169 concerns regarding the experiments and the main scientific contributions of this paper.

170 Response to R3:

- 171 • A main concern I have is that clinician performance is measured from only one expert. This limits the validity of the
 172 main claim of the paper that LLMs perform similar or better than clinical experts.

173 We now added a second evaluation by a clinical expert. This expert was chosen based on a voluntary positive response
 174 to an initial request that was sent out to 5 clinicians working in the field of neurology/epileptology. The evaluators
 175 have the equivalence to a neurology board certification (in Germany), and at least 1 year of working experience in an
 176 Epilepsy Monitoring Unit evaluating seizure behavior from video-recordings. The second clinician performed similar
 177 to the first clinician (*Clinician 1*: mean F1 score: 48.77, 95% CI: [48.53, 49.02] — *Clinician 2*: mean F1 score: 46.75,
 178 95% CI: [46.51, 46.99]), overall not changing our main findings. These results are now included in **Figure 2** of the
 179 manuscript and the rebuttal.

- 180 • The goal of this paper is unclear. Is it that the authors are trying to validate an LLM-based tool for a particular medical
 181 task or is it that they are trying to comprehensively assess LLMs and make general conclusions about their use in
 182 epilepsy? In either scenario, the experiments are not substantial enough for the claims made. If it is the former, we
 183 would need more extensive baselines and if it is the latter, we would need more relevant LLMs, potentially ones that are
 184 medical-specific.

185 The main goal of our study was to assess the applicability of LLMs for the extraction of diagnostic information from
 186 unstructured symptom descriptions, as those are often the only available source of information, particularly in the
 187 first-time patient visits. We chose a core diagnostic task in epilepsy as a representative case, but our analysis is, in
 188 principle applicable to other domains (refer Figure 3). We evaluated correctness, confidence, calibration, and—most
 189 importantly—a reasoning assessment, which offers insights not attainable with traditional ML methods. Our analysis
 190 further allowed us to examine the effects of experimental manipulations such as prompting strategies and impersonation,
 191 without requiring post-hoc retraining. Such strategies are not applicable to conventional ML classifiers. Accordingly,
 192 this study focuses on LLMs as a versatile tool that can be used by both non-experts and experts. In response to
 193 reviewer’s points, we have extended our analysis to include ML baselines as well as two additional medical LLMs and,
 194 as proof-of-concept, provide preliminary analysis in a different medical domain.

- 195 • The LLMs considered in this work are general purpose LLMs and are not specific to medical text. There are many
 196 medical-specific LLMs that are more suitable for this task, Med-Llama, Med42, Med-PaLM, Phi-4, etc.

197 We initially focused on general-purpose LLMs as they are widely accessible, instruction-tuned for broad task following,
 198 and commonly used by non-domain experts. Prior work shows that even without medical-specific training, general
 199 LLMs encode substantial clinical knowledge and perform well across healthcare tasks [11, 8], supporting their
 200 practical clinical utility, especially where domain-specific models are not readily available. **At the same time, we**
 201 **initially excluded medical-specific LLMs because the early generation of such models suffered from important**
 202 **limitations, including poor adherence to specialized instructions, prompt sensitivity, and restricted context length**
 203 (see Methods section lines 761-766 for more details). These drawbacks limited their robustness and applicability for
 204 our task. However, with the recent release of stronger medical instruction-tuned models, **we have now incorporated**
 205 **two models — OpenBioLLM-70B and Med42-70B** (fine-tuned on LLaMA-3 70B)—into Figure 2 in the rebuttal
 206 and the manuscript. *Please note, Med-PaLM weights are not publicly available in our region and Phi-4 is a general*
 207 *purpose model like GPT-4 and thus they are not included in this analysis.*

208 In some prompting conditions, these medical-specific models performed well, but their results varied substantially
 209 across formats, reflecting both overconfidence and calibration issues. OpenBioLLM-70B (mean F1: 48.09) achieved
 210 good scores in certain settings — particularly Chain-of-Thought (52.19) and self-consistency (53.06, close to GPT-
 211 4.0’s 53.44), but showed weaker zero-shot performance (38.56). Med42-70B (mean F1: 43.11) demonstrated better
 212 calibration than OpenBioLLM-70B, yet its overall F1 scores were lower (e.g., 31.09 in Few-Shot, 47.63 in self-
 213 consistency). While both medical LLMs approached general LLMs in specific cases, neither consistently matched
 214

Figure 2. Comparison of LLMs (6 general, **2 medical**) and impact of prompt engineering strategies [Zero-Shot (ZS), Few-Shot (FS), ZS-Chain-of-Thought (CoT), FS-CoT and Self Consistency (SC)] on performance, confidence and calibration. (a) Mean F1 scores for all models obtained by bootstrapping. The boxplot highlights a significant improvement with advanced prompt styles, showing performance comparable to clinicians and at par with naive classifier (F1 score of 38.2% (red dashed line) (b) Confidence scores improve consistently with in-context learning, with FS and FS-CoT demonstrating the highest gains. Confidence score=1/0 for green/red, respectively. (c) Calibration (Brier Score Loss, BCL) for each model and prompt style, with FS-CoT and SC showing the best calibration (least BCL). (d) Multidimensional performance visualization comparing model correctness, confidence, and calibration metrics, with solid lines representing the best-performing models

the performance of top general-purpose models such as GPT-4.0 (mean 52.10) or Mixtral-8x7B (mean 50.58) across formats. These updated results are included in the main manuscript (page 3, lines 139–145).

- *The two baselines considered are not fair baselines. As far as I understand, the dataset is categorical and therefore I believe that many machine learning models can be trained using this data. It will be useful to understand how classical ML models perform on this task.*

We have now included four standard ML classifiers (Naive Bayes, Support Vector Machine (SVM), K Nearest Neighbors (KNN), and Decision Tree) using embedding outputs derived from semantic NLP models (BERT [4], trained on general text corpora, and ClinicalBERT [6] trained on medical texts such as doctors’ notes and clinical reports). The results are presented in **Figure 4** in the rebuttal here and **Extended Data Figure 6** in the manuscript.

We found that GPT-4 achieved comparable or better performance than most pretrained-embedding ML pipelines, while requiring no model training or hyperparameter search; only the Decision Tree marginally surpassed GPT-4 in our setup. Classical ML models using BERT/ClinicalBERT embeddings showed a wide performance spread, with Naive Bayes < SVM < KNN < Decision Tree. This highlights the importance of model choice, which often outweighed the effect of the specific embedding. Across 10 folds, Decision Tree (DT) was the only classical model to exceed GPT-4 zero-shot (DT mean F1 = 0.576 ± 0.011 SE vs GPT-4 ZS 0.525 ± 0.009 SE). GPT-4 self-consistency was on par with the stronger ML baselines, while clearly outperforming weaker ones (e.g., Naive Bayes, SVM). These trends are also reflected in

Figure 3. Proof of Concept - Adaptation of the SemioLLM framework to the dermatology domain: *Top Left*: Only a small set of domain-specific variables (e.g., disease list, clinical expert) are changed in the prompt template to enable new medical use cases. *Top Right*: Example system and user prompt structure for symptom-based likelihood estimation. *Bottom Left*: Representative model output from GPT-4.0, parsed according to the defined schema, showing reasoning, sources, and estimated likelihood for each disease. *Bottom Right*: Model performance across different personas, with higher accuracy for domain expertise (dermatologist, 55%), and further improvement to 70% (dark green) using chain-of-thought (CoT) prompting

the significance marks in the figure.

Finally, we think that our study’s primary contribution lies not in maximizing classification performance, but in demonstrating that LLMs can provide clinically coherent explanations alongside competitive predictions. LLMs in our study were not trained on the target data yet produced competitive results with human-readable reasoning, confidence estimates, and direct applicability in clinical workflows without feature engineering. In clinical settings, the ability to understand and verify an AI system’s reasoning is of different value than accuracy gains from ML classifiers. Moreover, classical ML models require large annotated datasets, extensive task-specific feature engineering, and retraining for domain or language adaptation. In contrast, LLMs can flexibly adapt their explanations to different audiences (e.g., specialists vs. patients) and languages with minimal additional effort.

We have commented on our results in the manuscript on page 22 (lines 1174 - 1197).

- *Broadly, this paper appears to be simply an evaluation of general LLMs on a publicly available data and does not generate any data or clinical knowledge. For example, if the authors could explain why in some cases the LLMs are better than clinicians and evaluate whether those explanations can improve the current clinical knowledge, that would*

Figure 4. Comparison of standard Machine Learning (ML) Algorithms for SOZ localization task: (a) the weighted F1 scores k-folds (N=10) for classical ML models (Naive Bayes, Support Vector Machine (SVM), K Nearest Neighbors (KNN), Decision Tree) using BERT and ClinicalBERT embeddings as features in comparison to performance of GPT-4.0 (prompt strategy: self-consistency (SC) and zero-shot (ZS)). **** denotes p -value $< 1.00e-04$, * denotes p -values < 0.05 i.e. significant difference, ns denote no significant difference

significantly enhance the value of this paper.

Indeed the goal of the paper was to thoroughly evaluate performance of LLMs (and inspired by the reviewer’s comment, we now include medical LLMs’) in realistic, unstructured clinical narratives—a domain underexplored in previous work [9, 14, 15]. We quantify predictive accuracy, but also model confidence, calibration, and the validity of reasoning pathways through clinical evaluation. We identify both knowledge gaps and unexpected model strengths. Our focus lies in the applicability of novel LLM architectures for clinical diagnostics and is a step for testing the safety and clinically relevant use of foundation models in neurology and real-world digital medicine [16].

- *Minor comments: (1). Figure 3 caption: there are two (b)’s. (2). In figure 4c caption, there is a discrepancy between the x-axis tick label ‘English’ and the caption saying ‘Same English’.*

Thank you for noting these issues. We have corrected the duplicate (b) in Figure 3 and resolved the discrepancy in the Figure 4c caption in the revised manuscript.

References

- [1] Alim-Marvasti, A., Romagnoli, et al. Probabilistic landscape of seizure semiology localizing values. *Brain Communications*, 4(3):fcac130, 2022.
- [2] Brown, T., Mann, B., Ryder, N., Subbiah, M., Kaplan, J. D., Dhariwal, P., Neelakantan, A., Shyam, P., Sastry, G., Askell, A., et al. Language models are few-shot learners. *Advances in neural information processing systems*, 33: 1877–1901, 2020.
- [3] Christophe, C., Kanithi, P., Raha, T., et al. Med42-v2: A suite of clinical llms. 2024.
- [4] Devlin, J., Chang, M.-W., Lee, K., and Toutanova, K. Bert: Pre-training of deep bidirectional transformers for language understanding. In *Proceedings of the 2019 conference of the North American chapter of the association for computational linguistics: human language technologies, volume 1 (long and short papers)*, pp. 4171–4186, 2019.
- [5] Fonferko-Shadrach, B., Strafford, H., Jones, C., Khan, R. A., Brown, S., Edwards, J., Hawken, J., Shrimpton, L. E., White, C. P., Powell, R., et al. Annotation of epilepsy clinic letters for natural language processing. *Journal of Biomedical Semantics*, 15(1):17, 2024.

- 385 [6] Huang, K., Altosaar, J., and Ranganath, R. Clinicalbert: Modeling clinical notes and predicting hospital readmission.
386 *arXiv preprint arXiv:1904.05342*, 2019.
- 387 [7] Jang, Y., Kim, J., and Ahn, S. Structural reasoning improves molecular understanding of llm. *arXiv preprint*
388 *arXiv:2410.05610*, 2024.
- 389 [8] Kung, T. H., Cheatham, M., Medenilla, A., Sillos, C., De Leon, L., Elepaño, C., Madriaga, M., Aggabao, R., Diaz-
390 Candido, G., Maningo, J., et al. Performance of chatgpt on usmle: potential for ai-assisted medical education using
391 large language models. *PLoS digital health*, 2(2):e0000198, 2023.
- 392 [9] Li, J., Dada, A., Puladi, B., Kleesiek, J., and Egger, J. Chatgpt in healthcare: a taxonomy and systematic review.
393 *Computer Methods and Programs in Biomedicine*, 245:108013, 2024.
- 394 [10] Mora, S., Turrisi, R., Chiarella, L., Consales, A., Tassi, L., Mai, R., Nobili, L., Barla, A., and Arnulfo, G. Nlp-based
395 tools for localization of the epileptogenic zone in patients with drug-resistant focal epilepsy. *Scientific Reports*, 14(1):
396 2349, 2024.
- 397 [11] Nori, H., King, N., McKinney, S. M., Carignan, D., and Horvitz, E. Capabilities of gpt-4 on medical challenge
398 problems. *arXiv preprint arXiv:2303.13375*, 2023.
- 399 [12] Pal, A., Umapathi, L. K., and Sankarasubbu, M. Medmcqa: A large-scale multi-subject multi-choice dataset for
400 medical domain question answering. In *Conference on health, inference, and learning*, pp. 248–260. PMLR, 2022.
- 401 [13] Rueda, A., Hassan, M. S., Perivolaris, A., Teferra, B. G., Samavi, R., Rambhatla, S., Wu, Y., Zhang, Y., Cao, B.,
402 Sharma, D., et al. Understanding llm scientific reasoning through promptings and model’s explanation on the answers.
403 *arXiv preprint arXiv:2505.01482*, 2025.
- 404 [14] Singhal, K., Azizi, S., Tu, T., Mahdavi, S. S., Wei, J., Chung, H. W., Scales, N., Tanwani, A., Cole-Lewis, H., Pfohl, S.,
405 et al. Large language models encode clinical knowledge. *Nature*, 620(7972):172–180, 2023.
- 406 [15] Singhal, K., Tu, T., Gottweis, J., Sayres, R., Wulczyn, E., Hou, L., Clark, K., Pfohl, S., Cole-Lewis, H., Neal, D., et al.
407 Towards expert-level medical question answering with large language models. *arXiv preprint arXiv:2305.09617*, 2023.
- 408 [16] van Diessen, E., van Amerongen, R. A., Zijlmans, M., and Otte, W. M. Potential merits and flaws of large language
409 models in epilepsy care: A critical review. *Epilepsia*, 2024.
- 410 [17] Wei, J., Wang, X., Schuurmans, D., Bosma, M., Xia, F., Chi, E., Le, Q. V., Zhou, D., et al. Chain-of-thought prompting
411 elicits reasoning in large language models. *Advances in neural information processing systems*, 35:24824–24837,
412 2022.
- 413
414
415
416
417
418
419
420
421
422
423
424
425
426
427
428
429
430
431
432
433
434
435
436
437
438
439

Rebuttal

SemioLLM: Evaluating Large Language Models for Diagnostic Reasoning from Unstructured Clinical Narratives in Epilepsy

Meghal Dani^{1,2} Muthu Jeyanthi Prakash¹ Filip Rosa³ Zeynep Akata^{2,4} Stefanie Liebe^{3,5}

We sincerely thank all three reviewers for their insightful and positive feedback on our revised manuscript. We are grateful that the substantial revisions and new analyses conducted in response to the previous round of reviews have satisfactorily addressed all concerns. We believe that our updates have significantly strengthened the manuscript. In this revision, we have added clinician qualifications as requested and respond to reviewer's comment below.

R1: Review *The manuscript has improved significantly and now presents a thoughtful, rigorous evaluation of LLMs for semiology-to-SOZ mapping. Thank you for the detailed rebuttal and the substantial revisions made. The authors have clearly invested considerable effort in addressing the reviewers' concerns, and the updates are well executed. I have no further comments.*

Response to R1: We sincerely thank the reviewer for their positive evaluation of our revised manuscript and for acknowledging the rigor of our analyses and the effort put into the revisions. Their insightful comments led to significant improvements including cross-domain validation (dermatology, pulmonology), expanded reasoning analysis with systematic random sampling (n=81), and explicit discussion of dataset and demographic limitations. We are pleased that the updated version have fully addressed the concerns raised and questions asked.

R2: Review *Overall much improved. Minor: please specify what is the training qualifications of "clinician 1" and "clinician 2" since a significant claim is that the AI tool is on par with them and please double check if 2 sig. figs after the decimal are really needed for percentages quoted*

Response to R2: We thank the reviewer for the positive assessment and for these helpful suggestions. Both clinicians are neurologists at a university hospital in Germany with a specific focus and training in epilepsy care, including 1 year experience in exclusively treating patients in an Epilepsy Monitoring Unit ("clinician 1": equivalent in level to a board-certified neurologist and "clinician 2": a board-certified neurologist). We have added these details to the Methods section of the manuscript.

Regarding the number of significant figures, we have carefully re-examined all reported percentages and opted to retain two decimal places for percentages and other statistics following APA-style guidance [1, 2].

R3: Review *I thank the authors for the extensive work they did to address my comments. I have no further comments.*

Response to R3: We sincerely thank the reviewer for their rigorous feedback in the previous round, which led to substantial improvements including the addition of a second clinician evaluator (n=1,269 semiologies), two medical-specific LLMs, classical ML baselines, and cross-domain validation experiments. We are pleased that the current version satisfactorily addresses all prior concerns.

References

- [1] Association, A. P. *Publication Manual of the American Psychological Association*. American Psychological Association, Washington, DC, 7 edition, 2019.

¹University of Tübingen, Tübingen, Germany ²Helmholtz Munich, Munich, Germany ³Dept. of Neurology and Epileptology, University Clinic Tübingen, Hertie Institute for Clinical Brain Research, Tübingen, Germany ⁴Technical University of Munich, Munich, Germany ⁵Machine Learning in Science, Excellence Cluster Machine Learning, Tübingen University, Germany. Correspondence to: Meghal Dani <meghal.dani@uni-tuebingen.de>, Stefanie Liebe <stefanie.liebe@uni-tuebingen.de>.

055 [2] Cousineau, D. How many decimals? rounding descriptive and inferential statistics. *Journal of Mathematical Psychology*,
056 96:102368, 2020.
057
058
059
060
061
062
063
064
065
066
067
068
069
070
071
072
073
074
075
076
077
078
079
080
081
082
083
084
085
086
087
088
089
090
091
092
093
094
095
096
097
098
099
100
101
102
103
104
105
106
107
108
109

Rebuttal

SemioLLM: Evaluating Large Language Models for Diagnostic Reasoning from Unstructured Clinical Narratives in Epilepsy

Meghal Dani^{1,2} Muthu Jeyanthi Prakash¹ Filip Rosa³ Zeynep Akata^{2,4} Stefanie Liebe^{3,5}

We sincerely thank all three reviewers for their insightful and positive feedback on our revised manuscript. We are grateful that the substantial revisions and new analyses conducted in response to the previous round of reviews have satisfactorily addressed all concerns. We believe that our updates have significantly strengthened the manuscript. In this revision, we have added clinician qualifications as requested and respond to reviewer's comment below.

R1: Review *The manuscript has improved significantly and now presents a thoughtful, rigorous evaluation of LLMs for semiology-to-SOZ mapping. Thank you for the detailed rebuttal and the substantial revisions made. The authors have clearly invested considerable effort in addressing the reviewers' concerns, and the updates are well executed. I have no further comments.*

Response to R1: We sincerely thank the reviewer for their positive evaluation of our revised manuscript and for acknowledging the rigor of our analyses and the effort put into the revisions. Their insightful comments led to significant improvements including cross-domain validation (dermatology, pulmonology), expanded reasoning analysis with systematic random sampling (n=81), and explicit discussion of dataset and demographic limitations. We are pleased that the updated version have fully addressed the concerns raised and questions asked.

R2: Review *Overall much improved. Minor: please specify what is the training qualifications of "clinician 1" and "clinician 2" since a significant claim is that the AI tool is on par with them and please double check if 2 sig. figs after the decimal are really needed for percentages quoted*

Response to R2: We thank the reviewer for the positive assessment and for these helpful suggestions. Both clinicians are neurologists at a university hospital in Germany with a specific focus and training in epilepsy care, including 1 year experience in exclusively treating patients in an Epilepsy Monitoring Unit ("clinician 1": equivalent in level to a board-certified neurologist and "clinician 2": a board-certified neurologist). We have added these details to the Methods section of the manuscript.

Regarding the number of significant figures, we have carefully re-examined all reported percentages and opted to retain two decimal places for percentages and other statistics following APA-style guidance [1, 2].

R3: Review *I thank the authors for the extensive work they did to address my comments. I have no further comments.*

Response to R3: We sincerely thank the reviewer for their rigorous feedback in the previous round, which led to substantial improvements including the addition of a second clinician evaluator (n=1,269 semiologies), two medical-specific LLMs, classical ML baselines, and cross-domain validation experiments. We are pleased that the current version satisfactorily addresses all prior concerns.

References

- [1] Association, A. P. *Publication Manual of the American Psychological Association*. American Psychological Association, Washington, DC, 7 edition, 2019.

¹University of Tübingen, Tübingen, Germany ²Helmholtz Munich, Munich, Germany ³Dept. of Neurology and Epileptology, University Clinic Tübingen, Hertie Institute for Clinical Brain Research, Tübingen, Germany ⁴Technical University of Munich, Munich, Germany ⁵Machine Learning in Science, Excellence Cluster Machine Learning, Tübingen University, Germany. Correspondence to: Meghal Dani <meghal.dani@uni-tuebingen.de>, Stefanie Liebe <stefanie.liebe@uni-tuebingen.de>.

055 [2] Cousineau, D. How many decimals? rounding descriptive and inferential statistics. *Journal of Mathematical Psychology*,
056 96:102368, 2020.
057
058
059
060
061
062
063
064
065
066
067
068
069
070
071
072
073
074
075
076
077
078
079
080
081
082
083
084
085
086
087
088
089
090
091
092
093
094
095
096
097
098
099
100
101
102
103
104
105
106
107
108
109